# A Special Phenotype of Aconidial *Aspergillus niger* SH2 and Its Mechanism of Formation via CRISPRi

**DOI:** 10.3390/jof8070679

**Published:** 2022-06-28

**Authors:** Le-Yi Yu, Lin-Xiang Li, Lin-Lin Yao, Jun-Wei Zheng, Bin Wang, Li Pan

**Affiliations:** School of Biology and Biological Engineering, South China University of Technology, Guangzhou Higher Education Mega Center, Guangzhou 510006, China; 201911008179@mail.scut.edu.com (L.-Y.Y.); linxiang0214@163.com (L.-X.L.); linlinyaozsyx@163.com (L.-L.Y.); 201710106321@mail.scut.edu.com (J.-W.Z.); btbinwang@scut.edu.cn (B.W.)

**Keywords:** *Aspergillus niger* SH2, CRISPRi, morphology, N-acetyl-D-glucosamine, spore-like propagules

## Abstract

The complex morphological structure of *Aspergillus niger* influences its production of proteins, metabolites, etc., making the genetic manipulation and clonal purification of this species increasingly difficult, especially in aconidial *Aspergillus niger*. In this study, we found that N-acetyl-D-glucosamine (GlcNAc) could induce the formation of spore-like propagules in the aconidial *Aspergillus niger* SH2 strain. The spore-like propagules possessed life activities such as drug resistance, genetic transformation, and germination. Transcriptomic analysis indicated that the spore-like propagules were resting conidia entering dormancy and becoming more tolerant to environmental stresses. The *Dac1* gene and the metabolic pathway of GlcNAc converted to glycolysis are related to the formation of the spore-like propagules, as evidenced by the CRISPRi system, qPCR, and semi-quantitative RT-PCR. Moreover, a method based on the CRISPR-Cas9 tool to rapidly recycle screening tags and recover genes was suitable for *Aspergillus niger* SH2. To sum up, this suggests that the spore-like propagules are resting conidia and the mechanism of their formation is the metabolic pathway of GlcNAc converted to glycolysis, particularly the *Dac1* gene. This study can improve our understanding of the critical factors involved in mechanisms of phenotypic change and provides a good model for researching phenotypic change in filamentous fungi.

## 1. Introduction

Fungi are one of the most used species in the industry for enzyme production [1,2]. The filamentous fungus *Aspergillus niger* is widely used as a cellular factory to produce native or heterologous proteins, organic acids, drugs, etc. [1,3], due to its secretion efficiency, high-production capacity, capability of carrying out posttranslational modifications, and metabolic diversity [4,5]. However, the high productivity of the *Aspergillus niger* is strongly related to its specific morphological form [6,7]. For instance, in macroscopic morphology, pellet growth is beneficial to the production of citric acid and itaconic acid [8]. In other cases, freely dispersed mycelium is found to be optimal for the production of amylase, phytase, and neo-fucosyltransferase [9,10]. At the microscopic morphology level, *Aspergillus niger* grows as highly polarized tubular cells called hyphae [11]. The characteristic of hypha is branch growth at the tip and further extension [12]. In *Aspergillus niger*, the productions, e.g., proteins, principally appear in the tip of the hyphae [12,13]. Moreover, a study has already indicated that aconidial *Aspergillus niger* is beneficial for protein biosynthesis and secretion [14]. *Aspergillus niger* SH2, an aconidial phenotype, is a strain with high industrial glucoamylase (and other native or heterologous proteins) production [15]. Therefore, figuring out the critical factor for phenotypic change in *Aspergillus niger* is very important.

In addition, compared with yeast, the genetics and tools for the genetic manipulation of *Aspergillus niger* are not well developed or comprehensive [16]. This is mainly due to the complex structure of one of the most prominent characteristics of the hypha [13,17]. A hypha can fuse with other hyphae to form a network called mycelium [18]. During the fermentation process, the macroscopic morphology of the hyphae displays freely dispersed mycelium, clumps, or pellets [9,18,19]. Moreover, the cell nucleus of the *Aspergillus niger* can move freely throughout the mycelial network via the diaphragm hole. This means that *Aspergillus niger* presents a multicellular nucleus phenotype [17]. Pure homokaryotic strains, such as the cells that comprise a single nucleus of filamentous fungi, can be obtained by inducing conidium production from the hyphae and then isolating single spores [20,21]. However, many industrially used filamentous fungal strains, such as *Aspergillus niger* SH2, are no longer producing conidia [15]. This appears to be particularly common in high-protein production strains [3]. It is difficult for the complex structure of aconidial *Aspergillus niger* SH2 to endure genetic manipulation, clonal screening, and clonal purification. A previous study found that high levels of germination could be induced by N-acetyl-D-glucosamine in plain water [22,23,24]. Therefore, there is a need to explore new methods for *Aspergillus niger* SH2 manipulation to obtain simple structures and pure homokaryotic strains.

The CRISPR-Cas9 system is a nuclease-based genome editing technology which is widely used in eukaryotes and prokaryotes [25,26]. It consists of the associated Cas9 nuclease and an engineered guide RNA. The Cas9 protein can be further modified to form new genome editing technologies, such as transcription repressor (CRISPRi) and activation (CRISPRa) [27,28,29]. The core component of CRISPRi (CRISPR interference) is dead Cas9 (dCas9) which originates from two-point mutations of Cas9 [30,31]. Román et al. used fusions between a dCas9 and specific repressors (*Nrg1*) or activators (*Gal4*) that resulted in the specific repression or activation of cytosolic catalase [32]. The fusion of dCas9 to a *Mxi1* repressor domain enhanced the transcriptional repression in *Candida albicans* [33]. An inducible single plasmid CRISPRi system for gene repression in yeast was created and used to analyze the fitness effects of gRNAs under 18 small molecule treatments of CRISPRi by repression rather than a complete loss of function [34]. Therefore, the CRISPRi system can be used to explore the essential genes for controlling phenotypic change in the *Aspergillus niger* SH2.

The aims of the present study were: (1) to explore chemical agents such as N-acetyl-D-glucosamine (GlcNAc) with the potential for inducing the production of pure homokaryotic strains (i.e., the spore-like propagule) from aconidial *Aspergillus niger* SH2; (2) to determine the life activities of homokaryotic strains (spore-like propagule), such as drug resistance, genetic transformation, and germination; (3) to form a CRISPRi system in aconidial *Aspergillus niger* SH2; (4) to analyze the effect of genes or pathways on mediating the spore-like propagule response to N-acetyl-D-glucosamine using transcriptomic analysis and the CRISPRi system; and (5) to determine the essential genes for controlling phenotypic change in *Aspergillus niger* SH2. This study was expected to improve our understanding of the critical factors involved in the mechanism for phenotypic change in *Aspergillus niger* SH2 and provide a fine model for researching phenotypic change in filamentous fungi.

## 2. Materials and Methods

### 2.1. Strains and Media

The strains used in this study are listed in Appendix A. The following media were used in this study: Luria-Bertani medium (LB) containing (g L^−1^) tryptone (10), yeast extract (5), and NaCl (10); Czapek-Dox medium (CD) containing (g L^−1^) glucose (20), KCl (2), MgSO_4_·7H_2_O (0.5), KH_2_PO_4_ (1), NaNO_3_ (3), FeSO_4_·7H_2_O (0.01), pH 5.5; Hyper osmosis CD medium (HCD) containing (g L^−1^) sucrose (342.3), KCl (2), MgSO_4_·7H_2_O (0.5), KH_2_PO_4_ (1), NaNO_3_ (3), FeSO_4_·7H_2_O (0.01), pH 5.5; N-acetyl-D-glucosamine Czapek-Dox medium (NCD) containing (g L^−1^) N-acetyl-D-glucosamine (20), KCl (2), MgSO_4_·7H_2_O (0.5), KH_2_PO_4_ (1), NaNO_3_ (3), FeSO_4_·7H_2_O (0.01), pH 5.5; N-acetyl-D-glucosamine and glucose Czapek-Dox medium (NGCD) containing (g L^−1^) N-acetyl-D-glucosamine (20), glucose (20), KCl (2), MgSO_4_·7H_2_O (0.5), KH_2_PO_4_ (1), NaNO_3_ (3), FeSO_4_·7H_2_O (0.01), pH 5.5; Dextrose peptone yeast extract medium (DPY) containing (g L^−1^) glucose (20), tryptone (10), yeast extract (5), KH_2_PO_4_ (5), MgSO_4_·7H_2_O (0.5); Potato dextrose agar medium (PDA) containing (g L^−1^) potato extract powder (300), glucose (20), agar (2), chloramphenicol (0.1). The solid medium and semi-solid medium contained 2% (w:v) and 0.5% (w:v) agar, respectively. All materials were sterilized by autoclaving at 115 °C for 20 min before use.

### 2.2. The Formation of the Spore-like Propagule Assay in the Sporogenous Aspergillus niger SH2

The formation of the spore-like propagule (SPL) was induced by N-acetyl-D-glucosamine (GlcNAc) [17]. Firstly, the sporogenous *Aspergillus niger* SH2 strains were grown on a CD liquid medium in a biochemical incubator at 30 °C for 4 d. The hyphae were collected and washed thrice with sterile ddH_2_O, and milled by a low-speed TGrinder according to the user’s manual (TIANGEN Biotech (BEIJING) Co., Ltd., Beijing City, China) and diluted to a final OD_600 nm_ of 1.0. Next, 10 μL of the smooth hyphae was dropped onto NCD and CD solid medium, meanwhile, 100 μL of the smooth hyphae was dropped into NCD and CD liquid medium, and all mediums were incubated at 30 °C for 5 d. After incubation, the morphology of the strains was observed by Zeiss PrimoStar biological microscope (Carl Zeiss (SUZHOU) Co., Ltd., Suzhou City, Jiangsu Province, China).

### 2.3. The Purification of the Spore-like Propagule

To obtain the pure spore-like propagule for further research, the flowchart of spore-like propagule purification is shown in Appendix A. The first induced growth of spore-like propagule in NCD solid media was similar to the above information in 2.2. Firstly, the sporogenous *Aspergillus niger* SH2 strains were grown in CD liquid medium in a biochemical incubator at 30 °C for 4 d. The hyphae were collected and washed thrice with sterilized ddH_2_O and milled by a low-speed TGrinder according to the user’s manual (TIANGEN Biotech Co., Ltd., Beijing City, China) and diluted to a final OD_600 nm_ of 1.0. Next, 100 μL of the smooth hyphae was dropped onto an NCD solid medium and placed into a biochemical incubator at 30 °C for 5 d. After the first inducement of growth, the strains on the plate were collected and washed thrice with sterile ddH_2_O and resuspended in 10 mL of sterile ddH_2_O. The resuspension solution was filtered using a four-layer (Miracloth, Billerica, MA, USA) to remove the hyphae and collect the spore-like propagules. The spore-like propagules were then inoculated into an NCD liquid medium and placed in a biochemical incubator at 30 °C for 5 d. After the second inducement of growth, the fermentation broth was filtered using a four-layer (Miracloth, Billerica, MA, USA) to obtain the pure spore-like propagules.

### 2.4. The Routine and Fluorescent Staining of the Spore-like Propagules

The pure spore-like propagules were collected using the method above 2.3. The *Aspergillus niger* SH2 strains were grown in a CD liquid medium and placed in a biochemical incubator at 30 °C for 4 d to collect the hyphae. The *Aspergillus niger* CBS513.88 strains were dropped onto a PDA medium and placed in a biochemical incubator at 30 °C for 6 d to collect spores. The pure spore-like propagules, hyphae, and spores were stained by crystal violet and counterstained by safranine using a gram staining kit (Guangdong Huankai microbial Sci. and Tech. Co., Ltd., Guangzhou City, Gaungdong Province, China). Meanwhile, the pure spore-like propagules, hyphae, and spores were respectively stained by calcofluor white and DAPI. Finally, fluorescence images were captured by a Zeiss laser scanning confocal microscope 710.

### 2.5. Transmission Electron Microscopy

The spore-like propagules and hyphae were harvested and washed with sterilized ddH_2_O and subsequently fixed with 2.5% glutaraldehyde in 0.1 M pH 7.2 phosphate buffer overnight at 4 °C. The fixed samples were washed thrice with sterilized ddH_2_O and then they were dehydrated stepwise with methanol. The dehydrated samples were embedded in Epon 812 resin for sectioning. The ultrathin sections were subsequently stained with 2% uranium acetate and the resulting images were observed under a Tecnai Spirit 120 kV transmission electron microscope (FEI Co. of Hillsboro, OR, USA) [35]. Transmission electron microscopy was performed at Wuhan Service Technology Co., Ltd., Wuhan City, Hubei Province, China.

### 2.6. The Phenotypes and Genetic Transformation Assay of the Spore-like Propagule

For the phenotype assay of the spore-like propagule, 10 µL of the pure spore-like propagule and hypha suspension (OD_600 nm_ = 1.0) from *Aspergillus niger* SH2 were respectively inoculated on CD and NCD plates and grown for 7 d at 30 °C. The CD and NCD mediums, respectively, contained 300 µg/mL calcofluor white, 600 µg/mL cargo red, and 3 µM camptothecin. For the genetic transformation assay of the spore-like propagule, the plasmid pFC330 [36] was transferred into the pure spore-like propagules of *Aspergillus niger* SH2 ∆*Ku*∆*pyrG* using the protoplast transformation method [37]. The protoplast transformation method of *Aspergillus niger* is described below. Firstly, the pure spore-like propagules or hyphae were washed twice with sterilized ddH_2_O and 0.8 M NaCl solution, respectively. Protoplasts of the pure spore-like propagules or hyphae were prepared in an enzymatic lysis buffer composed of 0.8 M NaCl, 1% (W: V) cellulase, 1 (W: V) helicase, and 0.5% (W: V) lywallzyme. After enzymolysis, the enzymatic lysis buffer was filtered using a four-layer (Miracloth, Billerica, MA, USA) to collect the protoplasts. Then, the protoplasts were washed twice and resuspended with sterilized cold STC buffer (10 mM Tri-HCl, 1.2 M sorbitol, 50 mM CaCl_2_, pH7.5) for transformation. The transformation mixture was made up of 160 μL protoplasts, 100 μL plasmid solution, and 60 μL PEG buffer (60% PEG4000, 50 mM CaCl_2_, 10 mM Tri-HCl, pH7.5). The mixture was incubated on ice for 30 min and then another 1.5 mL PEG was added for 25 min incubation at room temperature. Finally, the mixture was pooled down on the HCD plate. After 5–7 days of cultivation at 30 °C, transformants were selected and identified. For the growth assay of the spore-like propagule, the pure spore-like propagules of *Aspergillus niger* SH2 were collected and then stored under different conditions (Figure 2C). After 12 months, the preserved spore-like propagules were recovered under a CD plate.

### 2.7. RNA Purification, Quantitative RT-PCR Analysis, and RNA-seq

The pure spore-like propagules or hyphae were collected and washed twice with sterilized ddH_2_O. The washed cells were ground to a fine powder in liquid nitrogen, and the total RNA was then extracted and purified using a HiPure Fungal RNA Kit (Magen Biotechnology Co., Ltd, Guangzhou City, Guangdong Province, China, China). Reverse transcription was performed using a PrimeScript RT-PCR Kit (TaKaRa Bio INC., Nojihigashi 7-4-38, Kusatsu, Shiga 525-0058, Japan) and a semi-quantitative RT-PCR method. The relative expression level of a selected gene was determined using the Power SYBR qPCR premix reagents (TransGen Biotech, Beijing City, China) in ABI 7500 Fast Real-Time PCR detection system. The relative transcript levels of the genes were calculated as a fold change and normalized to the housekeeping gene *gpdA* using the comparative 2^−ΔΔCT^ method [35]. The semi-quantitative RT-PCR system is shown in Appendix A. The primers used for qRT-PCR are listed in Appendix A. The RNA quality was assessed with NanoDrop and Agilent 2100 to confirm that all samples had an RNA integrity value. All samples were prepared in duplicate. Transcriptome library sequencing was performed by the Beijing Genomics Institute (BGI) with 50 bp single-end reads on the BGISEQ-500 (NCBI accession number, PRJNA588127). After the initial quality control, clean reads were mapped to the genome and cDNA using HISAT [38] and Bowtie2 [39]. The gene expression level was measured in TPM to determine unigenes using RSEM [40]. The differentially expressed genes were assessed using the DEseq2 v1.16.1 Bioconductor package and defined based on the fold change criterion (|log_2_(fold change)| > 1, *p* < 0.05) [35]. The FPKM value of all samples is provided in Appendix A. N1 and N2 in Appendix A mean the pure spore-like propagule samples in NCD, and G1 and G2 in Appendix A mean the hypha samples in CD.

### 2.8. Construction of CRISPRi in Aspergillus niger SH2

The CRISPRi system consisted of d*Cas9*-*Mxi1* and sgRNA. The d*Cas9*-*Mxi1* plasmid was constructed based on the pFC330 plasmid [36]. The sgRNA component was integrated into *aamA* (An11g03340) genetic locus in the *Aspergillus niger* SH2. The Cas9 expression cassette in the pFC330 plasmid was removed with *PacI* and *PmlI* restriction endonucleases digestion to generate the *Mxi1* vector for expression of dCas9 fusion proteins. The map of the recombinant pFC330-d*Cas9*-*Mxi1* plasmid is shown in Figure 4B and the sequence of the d*Cas9*-*Mxi1* is listed in the Appendix A. The sgRNA expression cassette consisted of an *A. fumigatus* U6 promoter, protospacer (20 bp), a sgRNA scaffold (76 bp), and an *A. oryzae* U6 terminator. Firstly, a common sgRNA plasmid without protospacer (20 bp) was constructed based on pMD18T (TaKaRa Bio INC., Nojihigashi 7-4-38, Kusatsu, Shiga 525-0058, Japan). The common sgRNA plasmid included the following: *aamA* upstream homologous arm, *A. fumigatus* U6 promoter, *Not I* restriction enzyme cutting site, sgRNA scaffold (76 bp), *A. oryzae* U6 terminator, *hygB* cassette, and an *aamA* downstream homologous arm. The sequence of common sgRNA cassettes is listed in Appendix A. The protospacer (20 bp) RNAs of the genes were predicted by CRISPR RGEN Tools (www.rgenome.net/cas-offinder/, accessed on 12 March 2021) and were evaluated according to the standards [41] by the Python 3.9. The high-score protospacer RNA sequence was transcribed by T7 in vitro Transcription Kit (New England Biolabs (Beijing) LTD., Beijing City, China). The transcription system and conditions are shown in Appendix A. After transcription, the product was recycled by isopropyl alcohol and washed twice with 75% icy ethanol, and then dried and resuspended with 50 μL RNase-free water. The ethanol precipitation system and conditions are shown in Appendix A. After recycling, cleavage of the target gene in vitro was measured by an NLS-Cas9 Nuclease Kit (Novoprotein Scientific Inc., Suzhou City, Jiangsu Province, China). The cleavage of the target gene system and condition are shown in Appendix A and the cleavage results were analyzed by gel electrophoresis. The flowchart of CRISPRi construction is shown in Appendix A. After sequencing analysis, the correct recombinant pFC330-d*Cas9*-*Mxi1* plasmid and linearized sgRNA vector were simultaneously transformed into the *A. niger* SH2 ∆*Ku*∆*pyrG* using the protoplast transformation method described in Section 2.6.

### 2.9. Construction of Gene Knockout or In-Situ Complementation Strains

The method of gene knockout was homologous recombination. To construct the *Dac1* et al. genes deletion plasmid, the *Dac1* gene upstream (ATG start codon) (1 kb) and downstream (TAA stop codon) (1 kb) regions and the *pyrG* gene (1398 bp) were amplified, respectively. The three fragments were connected to the pMD18T vector (TaKaRa Bio INC., Nojihigashi 7-4-38, Kusatsu, Shiga 525-0058, Japan) using a NEBuilder HiFi Assembly Kit (New England Biolabs (Beijing) LTD., Beijing City, China) to obtain the recombinant plasmid pMD18-D*Dac1*::*pyrG* (Figure 5A). Then, the linearized pMD18-D*Dac1*::*pyrG* plasmid was transformed into *A. niger* SH2 ∆*Ku*∆*pyrG* using the protoplast transformation method described in 2.6. After 5–7 days of cultivation at 30 °C, the transformants were selected and identified.

The method for gene in-situ complementation was the CRISPR-Cas9 tool and gene complement, fragment combining, phenotypic screening CD solid media contained 5-fluoroorotic acid, uridine, and *hygB* (Figure 5B). The CRISPR-Cas9 tool was constructed based on the pFC332 plasmid [36]. The *pyrG* (1398 bp) sgRNA expression cassette included: *A. fumigatus* U6 promoter, *pyrG* protospacer (20 bp), sgRNA scaffold (76 bp), and *A. oryzae* U6 terminator, and was cloned into the pFC332 plasmid using a NEBuilder HiFi Assembly Kit (New England Biolabs, America) to obtain the recombinant plasmid pFC332-*pyrG* sgRNA (Figure 5B). The gene complement fragment included: *Dac1* gene upstream (ATG start codon) (1 kb), *Dac1* gene (1365 bp), and downstream (TAA stop codon) (1 kb), and was cloned into the pMD18T vector (TaKaRa, Japan) using a NEBuilder HiFi Assembly Kit (New England Biolabs, America) to obtain the recombinant plasmid pMD18-R*Dac1* (Figure 5B). The sequence of the *pyrG* sgRNA expression cassette is listed in Appendix A. Then, the recombinant plasmid pFC332-*pyrG* sgRNA and the linearized pMD18-R*Dac1* plasmid were simultaneously transformed into the *A. niger* SH2 ∆*Dac1* using the protoplast transformation method described in Section 2.6. The screening method was performed using CD solid media supplied with 5-fluoroorotic acid and uridine. After 5–7 days of cultivation at 30 °C, transformants were selected and identified.

### 2.10. The Phenotypes Assay of Gene Knockout or In-Situ Complementation Strains

The sporogenous *Aspergillus niger* SH2 strain, *Dac1* gene knockout strain, and *Dac1* in-situ complementation strain were grown in a CD liquid medium and placed in a biochemical incubator at 30 °C for 4 d. The hyphae were collected and washed thrice with sterile ddH_2_O and milled by a low-speed TGrinder, according to the user’s manual (TIANGEN Biotech (BEIJING) Co., Ltd., Beijing City, China), and diluted to a final OD_600 nm_ of 1.0. Next, 10 μL of the smooth hyphae was dropped onto NCD, CD, and NGCD solid mediums, meanwhile 100 μL of the smooth hyphae was dropped into NCD, CD, and NGCD liquid mediums, and all mediums were incubated at 30 °C for 5 d. After incubation, the morphology of the strains was observed by Zeiss PrimoStar biological microscope (Carl Zeiss (SUZHOU) Co., Ltd., Suzhou City, Jiangsu Province, China).

### 2.11. Statistical Analyses

Each treatment in all experiments was conducted in triplicate. The data collected were statistically analyzed using Microsoft Excel 2016 and IBM SPSS Statistics 26.0. Data were subjected to analysis of variance (ANOVA), and the least significant differences (LSDs) were calculated using the *F*-method. All figures were made by Sigmaplot 12.0, and Adobe Photoshop CC 2019 was used to process the pictures.

## 3. Results

### 3.1. The Induction, Morphology, and Subcellular Structure Assay of the Spore-like Propagule in Sporogenous Aspergillus niger SH2

The strain *Aspergillus niger* SH2 does not produce spores in a liquid or solid medium. To induce the formation of spores in *Aspergillus niger* SH2, we inoculated the hyphae in various mediums containing different sole carbon sources, such as N-acetyl-D-glucosamine (GlcNAc), glucose, arabinose, xylose, and mannitol. Plenty of spore-like propagules were produced only in the liquid medium, and the solid medium contained N-acetyl-D-glucosamine (Figure 1A). The color of the fermentation broth added to N-acetyl-D-glucosamine made the colonies turn reddish-brown (Figure 1A). To determine the morphology and cellular structure of the spore-like propagules, we observed them under different stainings, such as crystal violet staining, safranine counterstaining, fluorescent white straining, and DAPI staining. Like spores, the spore-like propagules were ball-shaped (Figure 1B). They could be stained with crystal violet as blue, but it was more difficult to be counterstain them with safranine (as red) compared with the normal hyphae (Figure 1B). The spore-like propagules and normal hyphae could be stained with fluorescent white and DAPI (Figure 1C). Moreover, as with the normal hyphae, the spore-like propagules had abundant subcellular organelles, such as cell walls, vacuoles, plastosomes, nucleuses, endoplasmic reticulum, etc. (Figure 1D). The cell wall of the spore-like propagules had two films; the outer layer (darker part) and the inner layer (white part) (Figure 1D, E). The cell wall of the spore-like propagules (0.70 μm) was thicker than the normal hyphae (0.39 μm) (Figure 1D,E).

### 3.2. The Morphological Phenotype, Genetic Transformation, and Germination Assay of the Spore-like Propagule

To determine whether the spore-like propagule exhibited life activity, such as drug resistance, genetic transformation, and germination, we analyzed genetic transformation, germination, and morphological phenotype under calcofluor white, cargo red, and camptothecin of the spore-like propagules. The spore-like propagules grew well on the CD plates, and the NCD plates contained 300 µg/mL calcofluor white, 600 µg/mL cargo red, and 3 µM camptothecin (Figure 2A). The spore-like propagules and normal hyphae grew better on CD than on the NCD (Figure 2A). The spore-like propagules generated protoplasts using cellulase, helicase, and lywallzyme, however the enzymolysis time of the spore-like propagules (4 h) was longer than the normal hyphae (2 h) (Figure 2B). The pFC332 plasmids were transformed into the protoplasts of the spore-like propagules, with some transformations occurring on the plate (Figure 2C). The spore-like propagules were preserved in different conditions, such as sterile water, 30% glycerin, and saline at 4 °C or −20 °C (Figure 2C). After 12 months of preservation, the spore-like propagules were all well revived under different preservation conditions (Figure 2C).

### 3.3. The RNA Sequence of Spore-Like Propagule

To determine the effect of the genes or pathways for mediating the spore-like propagule’s response to N-acetyl-D-glucosamine, we performed transcriptomic analysis between the spore-like propagule (treatment) and the normal hypha (control). The results of the RNA sample quality detection were all qualified (Appendix A), and the correlation analysis of the RNA sequencing results was 0.998–1.000 (Appendix A). An amount of 11,954 genes were simultaneously detected in all samples (Appendix A). The gene ID, FPKM, log_2_(Fold Change), *p*-value, and description are shown in Appendix A. The RNA sequence results indicated that 8161 (4054 up-regulated genes and 4107 down-regulated genes) genes were differentially expressed in the spore-like propagule compared with the normal hypha (Figure 3A). After analysis (|log_2_(N/G)| > 1, *p* < 0.05) and filtering the data, the results indicated that 2449 (1059 up-regulated genes and 1390 down-regulated genes) genes were differentially expressed in the spore-like propagule compared with the normal hypha (Appendix A). We chose the 20 highest up-regulated genes and the 20 highest down-regulated genes between the samples for visible analysis (Figure 3B). The 20 highest up-regulated genes contained expressions of germinating conidia, mannitol dehydrogenase (*mtdA*), hydrophobins (*hfbA*, *hfbB*), glucosamine-6-phosphate deaminase, 1,3-beta-glucanosyltransferase (*bgt1*) (Figure 3B from Appendix A). The 20 highest down-regulated genes contained cellular responses to hypoxia and pathogenesis, Hsp90 chaperone (*sspB*), acid alpha-amylase (*aamA*), C-4 methyl sterol oxidase (*erg25*), alpha-amylase (*amyA*), mitochondria-localized alternative oxidase (*aox1*), Hsp70 protein gene, flavohemoglobin (*fhbA*), and mannitol 1-phosphate dehydrogenase (*mpdA*) (Figure 3B; Appendix A). Meanwhile, we analyzed the distribution of all the differentially expressed genes (Figure 3C). And we obtained the top 34 up-regulated genes and the top 59 down-regulated genes from Figure 3C (Appendix A). Then, we analyzed the 20 most significantly enriched categories of the genes using the GO method. After GO analysis, the significantly enriched categories of the top 34 up-regulated genes included: glucosamine catabolic process, N-acetylglucosamine catabolic process, N-acetylglucosamine-6-phosphate deacetylase activity, pathogenesis, intracellular, cellular response to N-acetyl-D-glucosamine, glucosamine-6-phosphate deaminase activity, inositol oxygenase activity, ion channel activity, ion transport, etc. (Figure 3D). The most significantly enriched categories of the top 59 down-regulated genes included: TRC complex, protein refolding, mRNA binding, putrescine biosynthetic process, ornithine decarboxylase activity, positive regulation endodeoxyribonuclease activity, glucan 1,4-al activity, ATPase inhibitor activity, protein unfolding, DNA biosynthetic process, polysaccharide metabolic process, etc. (Figure 3E).

Furthermore, the enriched pathways of 416 differentially expressed genes were mostly gathered on the MAPK-glucosidase signaling pathway (Figure 3F). The gene ID, log_2_(Fold Change), *p*-value, and description of 416 MAPK signaling pathway genes are shown in Appendix A. The results indicated that 194 up-regulated genes and 215 down-regulated genes were differentially expressed in the spore-like propagule compared with the normal hypha on the MAPK signaling pathway (Appendix A). We then analyzed the distribution of the 416 differentially expressed genes (Figure 3G). The top up-regulated genes of the MAPK signaling pathway contained: ATF/CREB family transcription factors, catalase activity, cellular aromatic compound metabolic processes, oxidoreductase activity, cellular responses to heat, cell cortex of cell tips, responses to oxidative stress, ferric iron binding, etc. (Figure 3G). The top down-regulated genes of the MAPK signaling pathway contained: hyphal growth, cellular bud tip, coenzyme binding, protein phosphorylation, transmembrane transport, cell septum, cellular metabolic process, etc. (Figure 3G)

Moreover, we found that some differentially expressed genes were related to the conidia-associated genes and germination-associated genes, and we found that the cell wall of the spore-like propagules was thicker than the normal hyphae (Figure 1D,E). Therefore, we analyzed the expression of conidia-associated genes (CAGs), germination-associated genes (GeAGs), and genes involved in the production of enzymes in cell wall synthesis or processing between the spore-like propagule and the normal hypha. The results indicated that all CAGs expression in the spore-like propagule was significantly higher than the genes expressed in the normal hypha (Figure 3H; Appendix A), especially the gene (An08g09880, *hfbD*) expressed in dormant conidia, which was greatly expressed in the spore-like propagule (Appendix A). However, the results indicated that only a small number of CAGs expressions in the spore-like propagule were higher than the genes expressed in the normal hypha (Figure 3I; Appendix A). The gene (An01g10790) expressed in the germinating conidia was greatly expressed in the spore-like propagule (Appendix A). However, the expression of another gene (An12g08230), also expressed in the germinating conidia, was lower in the spore-like propagule than in the normal hypha (Appendix A). For the genes involved in the production of enzymes in cell wall synthesis or processing, some genes related to spore-walls (An03g02360, An03g02400, An04g08500) were greatly expressed in the spore-like propagule (Figure 3J; Appendix A). Additionally, the 1,3-beta-glucanosyltransferase genes (An08g03580, An03g06220) were also greatly expressed in the spore-like propagule (Figure 3J; Appendix A). Interestingly, some chitin synthase genes (An02g02340, An02g02360, An06g01000) and beta-1,3-glucan synthase genes (An06g01550, An17g02120) had restrained expression in the spore-like propagule (Figure 3J; Appendix A). However, some chitin synthase genes (An09g04010, An12g10380) and alpha-1,3- glucan synthase genes (An09g03070) were greatly expressed in the spore-like propagule (Figure 3J; Appendix A).

### 3.4. The Construction of CRISPRi on Conidia and GlcNAc Metabolic Pathway Genes in Aspergillus niger SH2

Some essential genes were difficult to delete, so we constructed a CRISPRi system in *Aspergillus niger* SH2. The diagram of the CRISPRi working principle is shown in Figure 4B. Firstly, we analyzed the GlcNAc metabolic pathway, as shown in Figure 4A. The red font denotes the relevant genes. The transcriptomic analysis, gene ID, and gene description of GlcNAc metabolic pathway genes, and the differentially expressed genes related to conidia and MAPK signaling pathway, are shown in Appendix A The expression of GlcNAc metabolic pathway genes (*Ngt1*, *Hxk1*, *Dac1*, and *Uap1*) was greatly higher in the spore-like propagule than in the normal hypha (Appendix A). Then, we designed a spacer RNA system for the above-mentioned genes. The efficiency of the spacer RNA was verified in vitro by T7 transcription and Cas9 digestion systems, as shown in Appendix A. The results indicated that the whole design of the spacer RNA could work in vitro (Appendix A). The efficient sequence of the spacer RNA is listed in Appendix A Finally, the sequence of the spacer RNA was connected into a linearized sgRNA general vector using the infusion method. The flow chart of the spacer RNA assembly system is shown in Appendix A. The sequencing analysis indicated that the whole spacer RNA could be seamlessly connected into a linearized sgRNA general vector using the infusion method (Appendix A). The map of the CRISPRi system designed in this study is shown in Figure 4B. The CRISPRi system consisted of a d*Cas9*-*Mxi1* expression cassette and a sgRNA expression cassette. The d*Cas9*-*Mxi1* expression cassette was assembled into a pFC330 plasmid. The sgRNA expression cassette was inserted into the genome of the *Aspergillus niger* SH2 (*aamA* gene locus).

To find what key genes regulated and controlled the spore-like propagule formation, the different sgRNA CRISPRi systems were transferred into *Aspergillus niger* SH2. After selection and identification, the correct transformants were dropped onto NCD plates for phenotypic analysis. The results indicated that some transformants had different phenotypes compared with the *Aspergillus niger* SH2 wild strain, especially the *Dac1*-CRISPRi transformant (Figure 4C). The qPCR results indicated that most genes were restrained by the CRISPRi system, especially the *Dac1*, *Uap1*, and *hfbD* genes (Figure 4D). The semi-quantitative RT-PCR results indicated that the expression of *hfbD* (An08g09880), *Ngt1* (An16g09020) An12g00480, and *Hxk1* (An13g00510) reduced when the *Dac1* (An16g09040) gene was restrained by the CRISPRi system (Figure 5E). Meanwhile, when the *Dac1* (An16g09040) gene was restrained, the transformant grew slowly, and the colonies almost failed to become reddish-brown in color (Figure 5F). Importantly, there were few spore-like propagules in the NCD cultivation when the *Dac1* (An16g09040) gene was restrained by the CRISPRi system (Figure 5F).

### 3.5. The Phenotypes Assay of Dac1 Gene Knockout and In-Situ Complementation Mutants in Aspergillus niger SH2

To confirm that the *Dac1* gene could control spore-like propagule formation, we deleted and recovered the *Dac1* gene. After selection and identification, we obtained the mutants in which the *Dac1* gene had been deleted successfully (Figure 5A). Then, we structured a CRISPR-Cas9 tool so as to recycle the screening tag (*pyrG*) and recover the *Dac1* gene simultaneously (Figure 5B). The results indicated that the screening tag (*pyrG*) was recycled and, simultaneously, the *Dac1* gene recovered (Figure 5C). After identification, the strains, including the *Aspergillus niger*
*SH2* strain, the *Dac1* knockout mutant, and the *Dac1* recovery strain, were dropped onto an NCD medium for phenotypic analysis. The results indicated that the *Dac1* knockout mutant did not grow in the NCD medium, and grew slowly in the NGCD medium (Figure 5D). Meanwhile, the *Dac1* knockout mutant did not produce a spore-like propagule, and the color of the colony did not become reddish-brown in the NGCD medium (Figure 5D). On the contrary, the wild strain and *Dac1* recovery strain could produce spore-like propagules, and the color of the colony became reddish-brown in the NCD and NGCD mediums (Figure 5D).

## 4. Discussion

The *Aspergillus niger* SH2 has a distinct aconidial phenotype, which could be explained by the loss of the homolog of the *Aspergillus nidulans PrpA* gene (An18g01170) in the 200 kb deletion fragment [15]. The *PrpA* gene takes part in the formation of asexual sporulation [42]. The aconidial phenotype of the *Aspergillus niger* SH2 increases the difficulty of gaining homokaryons of desirable *Aspergillus niger* strains. This presents a significant obstacle when applying the workflow of genetic manipulation, clonal screening, and clonal purification. Oakland et al. reported (2021) a method for producing a homokaryotic derivative of the filamentous fungal cells, and they found that filamentous fungal cells could enable the spontaneous formation of spore-like propagules (SLPs) via the chemical agent N-acetyl-D-glucosamine [17]. However, they did not elucidate the cellular structure and formation mechanism of the spore-like propagule. In this study, the sporogenous *Aspergillus niger* SH2 wild strain produced a spore-like substance only in/on the mediums containing N-acetyl-D-glucosamine (Figure 1A), and we named this spore-like propagule (SLP) referring to Oakland et al. [17]. N-acetyl-D-glucosamine is known to be a potent inducer of morphological transition in dimorphic yeasts such as *Candida albicans* and *Yarrowia lipol* [43]. A previous study described a specific induction of chlamydospores in *Candida albicans* by N-acetyl-D-glucosamine, with high levels of germination induced by N-acetyl-D-glucosamine in plain water [22,23,24]. Other chemical agents, such as glucose, glucosamine, galactosamine, N-acetylgalactos-amine, fructose, and glucose-l-phosphate, were unable to replace N-acetyl-D-glucosamine in chlamydospore induction, which was stimulated only slightly by MgSO_4_ (0.1 mM) or MnCl_2_ (0.1 mM) [22,23]. Additionally, N-acetyl-D-glucosamine has been reported to facilitate yeast-to-filament conversion in thermally dimorphic fungi [44]. The spore-like propagule was more difficult to counterstain with safranine, and the cell walls of the spore-like propagules were thicker than the normal hyphae. Our results are consistent with the previous study, which reported that Gliousc-N-stimulated cell walls were approximately three times thicker than those of the cells grown in YPD or mating media. The cell walls in the majority of the cell population contained multiple layers of architecture upon GlcN induction [35]. Therefore, this suggests that N-acetyl-D-glucosamine is both the monomer to induce the formation of the spore-like propagules that possess a complete cellular structure and a specific activator of the enzyme for chitin synthetase in *Aspergillus niger* SH2.

Concerning the spore-like propagules, we analyzed genetic transformation, germination, and morphological phenotype under calcofluor white, cargo red, and camptothecin. Congo red can affect the β-1,3-glucan structure by restraining cell wall synthesis [45], while calcofluor white binds and disturbs cell wall chitin [46]. We found that the spore-like propagules had drug resistance on calcofluor white and cargo red. Moreover, preserving the sporogenous *Aspergillus niger* SH2 is complicated because the construction of the *Aspergillus niger* SH2 hypha is polymorphic, and it is hard to recover the hyphae after a long time of preservation. In this study, the spore-like propagules were all well revived even in sterile water and under room temperature conditions. This illustrates that the spore-like propagules represent a good agent for preservation. Despite the fact that the post-genome era of the filamentous fungi has begun, filamentous fungi such as *Aspergillus niger* have multicellular characteristics [47]. Compared with single-celled fungi, such as yeast, the growth and development of *Aspergillus niger* is relatively complex, making molecular genetic manipulation more difficult [13]. The spore-like propagules exhibit life activities. However, the cell wall of the spore-like propagule is too thick, meaning that the efficiency of transformation is low. However, it is important to note that obtaining a pure mutant is easy. Therefore, the spore-like propagules exhibit life activities and can be used as a good agent for preservation. Additionally, the genetic material can be transformed into spore-like propagules to obtain the pure mutant.

The RNA-seq analysis indicated that the enriched pathways of 416 differentially expressed genes were mostly gathered on the MAPK signaling pathway. The top up-regulated genes of the MAPK signaling pathway contained ATF/CREB family transcription factors, etc. Due to their ability to cope with various environments, *Candida* species can use multiple carbon sources, eventually triggering different signaling pathways, such as MAPK and cyclic AMP pathways, to adapt to environmental stresses and host niches [48,49,50]. ATF/CREB proteins can bind as homodimers or heterodimers to the specific DNA sequence T(G/T) ACGT (C/A) A through a bZIP structural motif, which consists of a leucine zipper that mediates dimerization, and an adjacent basic region that binds to DNA [51]. ATF/CREB transcription factors play important roles in the expression of osmotic stress-responsive genes that operate downstream of the MAP kinase pathway in the yeasts *Schizosaccharomyces pombe* and *Saccharomyces cerevisiae* [52,53]. The function of the *after* gene, belonging to the ATF/CREB family, was observed to have involvement in the stress tolerance of conidia in *Aspergillus oryzae* EST [54]. On the contrary, the highly down-regulated genes of the MAPK signaling pathway contained the following: oxidoreductase activity, muconate cycloisomerase, DNA binding, UDP-glucose 4-epimerase, secondary metabolite biosynthetic process, etc.. Additionally, the significantly enriched categories of the highly down-regulated genes were protein refolding, mRNA binding, ATPase inhibitor activity, protein unfolding, DNA biosynthetic processes, and polysaccharide metabolic processes. This indicates that metabolic activity is reduced in the spore-like propagule. Like conidia, the spore-like propagule has advanced abilities in stress tolerance [55].

Moreover, the 20 highest up-regulated genes were expressed in germinating conidia, and hydrophobins were expressed in dormant conidia. A previous study defined the genes that were dominantly expressed in conidia or germinating conidia as conidia-associated genes (CAGs) or germination-associated genes (GeAGs), respectively [56]. Therefore, we analyzed the expression of conidia-associated genes (CAGs) and germination-associated genes (GeAGs) in the spore-like propagules and the normal hyphae of *Aspergillus niger* SH2. We found that all of the CAGs in the spore-like propagule were significantly higher than the genes expressed in the normal hypha, but only a small number of CAGs expression in the spore-like propagule were higher than the genes expressed in the normal hypha. The GeAGs mainly included genes involved in fundamental cellular processes, such as ribosome biogenesis, nucleotide biogenesis, ubiquitin, and translation factors [56]. In the spore-like propagule, some fundamental cellular processes may stop. This means that the expression of the GeAGs is different in the spore-like propagule than it is in the germinating conidia. However, the expression of the GeAGs in the spore-like propagule is consistent with dormant conidia. This suggests that, like the dormant conidia, the spore-like propagules are resting conidia entering dormancy where they become more tolerant to environmental stresses.

On account of the thicker cell wall of the spore-like propagule, we analyzed the expression of the genes involved in the production of enzymes in cell wall synthesis or processing. Some chitin synthase genes (An09g04010, An12g10380) and an alpha-1,3- glucan synthase gene (An09g03070) were greatly expressed in the spore-like propagule. Further research of wall synthesis genes in spore-like propagules is needed. Moreover, the significantly enriched categories of the highly up-regulated genes included glucosamine catabolic processes, N-acetylglucosamine catabolic processes, N-acetylglucosamine-6-phosphate deacetylase activity, etc. These enriched categories are related to the N-acetylglucosamine (GlcNAc) metabolic process. N-acetylglucosamine (GlcNAc) is a monosaccharide signaling molecule that can regulate morphological transitions in *Candida albicans* and *Candida tropicalis* [57,58]. A proposed model of the GlcNAc catabolizing pathways in *C. Albicans* and *C. tropicalis* is shown in Figure 4A. After analysis, we found that the GlcNAc metabolic process genes, including *Ngt1* (An16g09020), *Hxk1* (An13g00510), and *Dac1* (An16g09040), were greatly expressed in the spore-like propagule (Appendix A). Therefore, for the next study, we will focus on conidia, the MAPK signaling pathway, and GlcNAc metabolic pathway genes.

Some essential genes were difficult to delete. For instance, the *agm1* (An18g05170) gene was not deleted in the *Aspergillus niger* SH2 of our previous study. CRISPRi provides a good alternative to other approaches for studying essential genes [59,60,61]. Smith et al. created an inducible single plasmid CRISPRi system for gene repression in yeast and used it to analyze the fitness effects of gRNAs under 18 small molecule treatments [34]. Correlating with the on-target activity of the Cas9, the sgRNA complex will enable a more effective application of CRISPR technology to edit the genome and probe gene functions [41]. In the CRISPRi system, the design criteria that maximize sgRNA efficacy are very important. Doench et al. quantitatively assayed the activities of thousands of sgRNAs to uncover the sequence features that modulate the ability of Cas9 to bind to DNA, cleave the target site, and produce a null allele [41]. Similar approaches were previously applied to RNAi knockdown [62,63]. We found that our spacer RNA system, which was designed using CRISPR RGEN Tools (www.rgenome.net/cas-offender/, accessed on 12 March 2021), standards [41], and Python 3.9, worked effectively for conidia and GlcNAc metabolic pathway genes in vitro and in vivo. This suggests that the CRISPRi system we designed can be used in *Aspergillus niger* SH2. Moreover, we found that there were few spore-like propagules were found in NCD cultivation and that the transformant grew slowly when the *Dac1* gene was restrained by the CRISPRi system. The *Dac1* gene can encode N-acetylglucosamine-6-phosphate deacetylase. Addionally, GlcNAc-6-PO4 is converted to fructose-6-PO4 in a stepwise manner by *Dac1* (GlcNAc-6-phosphate deacetylase) and *Nag1* (glucosamine-6-phosphate deaminase). Fructose-6-phosphate then enters mitochondria for glycolysis [57]. Therefore, the CRISPRi system we designed can be used in *Aspergillus niger* SH2, with the *Dac1* (An16g09040) gene and the metabolic pathway of GlcNAc converted to glycolysis being related to the formation of the spore-like propagules.

To confirm that the *Dac1* gene could control spore-like propagule formation, the *Dac1* gene was deleted and recovered. We found that the *Dac1* knockout mutant did not produce the spore-like propagules in an NGCD medium. In *Candida albicans*, which is an opportunistic pathogen, GlcNAc activates signal transduction for hyphal cell morphogenetic switching and virulence functions [64,65]. GlcNAc is also a potent inducer of hyphal cell morphology in two dimorphic fungi: *Histoplasma capsulatum* and *Blastomyces dermatitis* [44]. Similar to *Candida albicans* and other dimorphic fungi, the GlcNAc catabolic pathway genes in *Magnaporthe oryzae* are also organized in a genomic cluster [66,67]. It has been suggested that *Ngt1*, a GlcNAc transporter in *Candida albicans*, plays a crucial role in hyphal development during the response to GlcNAc, as deletion of *Candida tropicalis Ngt1* induces filamentous growth [68]. However, *Ngt1*-deleted *Candida albicans* mutants exhibit suppressed hyphal development [24]. There is little research to elucidate the mechanism of the hyphal cell morphology induced by GlcNAc. We found that the *Dac1* gene could control spore-like propagule formation. To sum up, this suggests that the *Dac1* (An16g09040) gene and the metabolic pathway of GlcNAc converted to glycolysis are related to the formation of the spore-like propagules in *Aspergillus niger* SH2.

In conclusion, N-acetyl-D-glucosamine (GlcNAc) is both the monomer to induce the formation of the spore-like propagules that possess a complete cellular structure and a specific activator of an enzyme for the chitin synthetase in *Aspergillus niger* SH2. The spore-like propagules exhibit life activities and can be used as a good agent for both preservation and genetic material to obtain the pure mutant. Like dormant conidia, the spore-like propagules are resting conidia entering dormancy and becoming more tolerant to environmental stresses. Importantly, the *Dac1* (An16g09040) gene and the metabolic pathway of GlcNAc converted to glycolysis are related to the formation of the spore-like propagules in *Aspergillus niger* SH2. Moreover, the CRISPRi system we designed, and the CRISPR-Cas9 tool used to rapidly recycle the screening tag (*pyrG*) and recover genes, can work efficiently in *Aspergillus niger* SH2.

## Figures and Tables

**Figure 1 jof-08-00679-f001:**
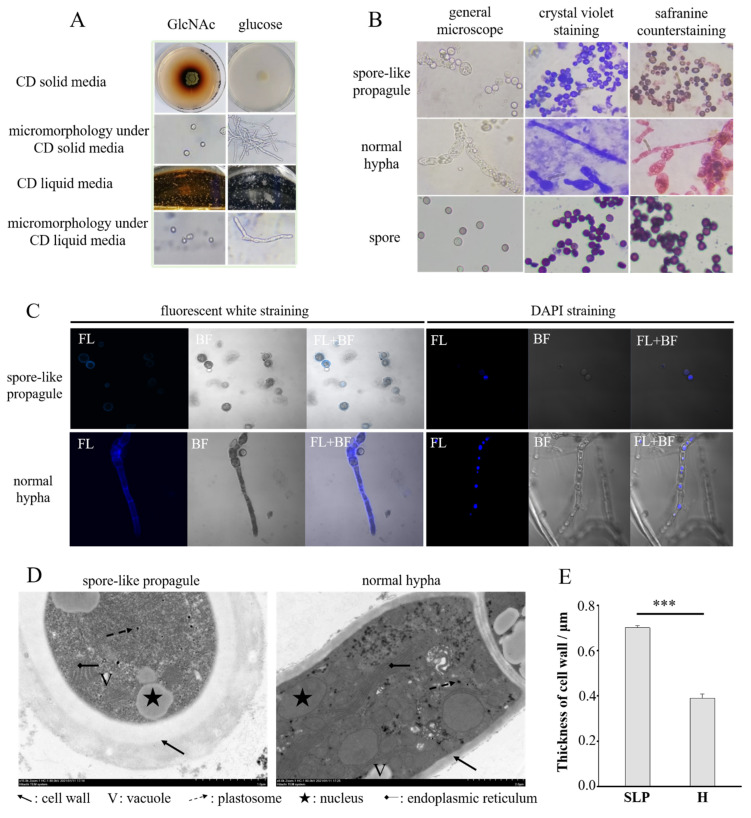
The morphology and subcellular structure assay of spore-like propagules in asporogenous *Aspergillus niger* SH2. (**A**) The induced growth of *Aspergillus niger* SH2 in NCD and CD. GlcNAc means NCD contains N-acetyl-D-glucosamine (GlcNAc), glucose means NCD contains glucose. An amount of 10 μL of the smooth hyphae was dropped onto NCD and CD solid medium, meanwhile 100 μL of the smooth hyphae was dropped into NCD and CD liquid medium, and all mediums were incubated at 30 °C for 5 d. (**B**) The morphology assay of the spore-like propagule, with normal hypha and spore as the control. The pure spore-like propagule, and hypha and spore, were stained by crystal violet and counterstained by safranine using a gram staining kit. The morphology was observed under general microscope, crystal violet staining and safranine counte staining using a Zeiss PrimoStar biological microscope. (**C**) The cell wall observation (by fluorescent white straining) and cell nucleus observation (by DAPI straining) of the spore-like propagule under a Zeiss laser scanning confocal microscope 710, normal hypha as the control. (**D**) TEM-based visualization of subcellular structure morphology of the spore-like propagule, normal hypha as the control. (**E**) The thickness of cell wall around the spore-like propagule, normal hypha as the control. The thickness data comes from the above TEM. SLP: the spore-like propagule, H: the normal hypha. The *** means *p* < 0.001.

**Figure 2 jof-08-00679-f002:**
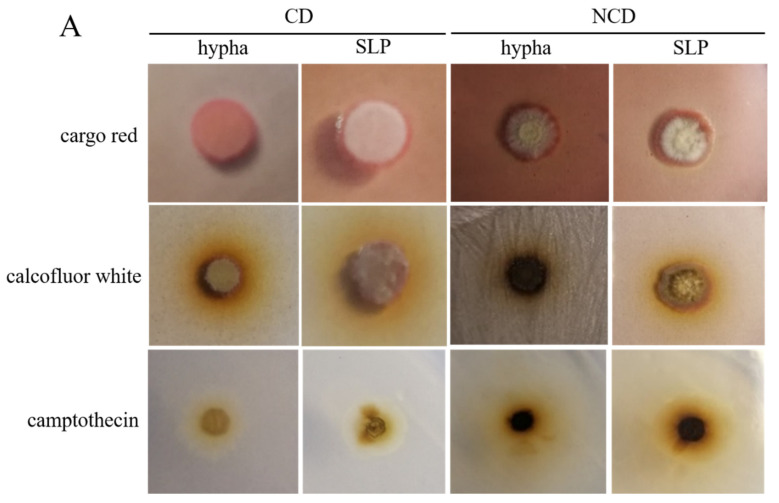
The life activity assay of the spore-like propagule. (**A**) The morphological phenotype of the spore-like propagules and normal hyphae on CD plates and NCD plates contained 300 μg/mL calcofluor white, 600 μg/mL cargo red, and 3 μM camptothecin. SLP means that the spore-like propagules were inoculated onto CD plates and NCD plates, and hypha means that the normal hyphae were inoculated onto CD plates and NCD plates. (**B**) The enzymolysis of spore-like propagules and chemical conversion of the protoplasts of the spore-like propagules for genetic transformation assay. The pFC330 plasmid was transferred into *A. niger* SH2 ∆*ku*∆*pyrG*. (**C**) The different preservation conditions of spore-like propagules and the recovery of spore-like propagules. The pure spore-like propagules were preserved in sterile water, 30% glycerin, and 0.8 M/0.4 M/0.2 M saline under room temperature, 4 °C and −80 °C, respectively. After 12 months, the preserved spore-like propagules were recovered under CD plates. The blue letters were the related preservation conditions and the corresponding plates.

**Figure 3 jof-08-00679-f003:**
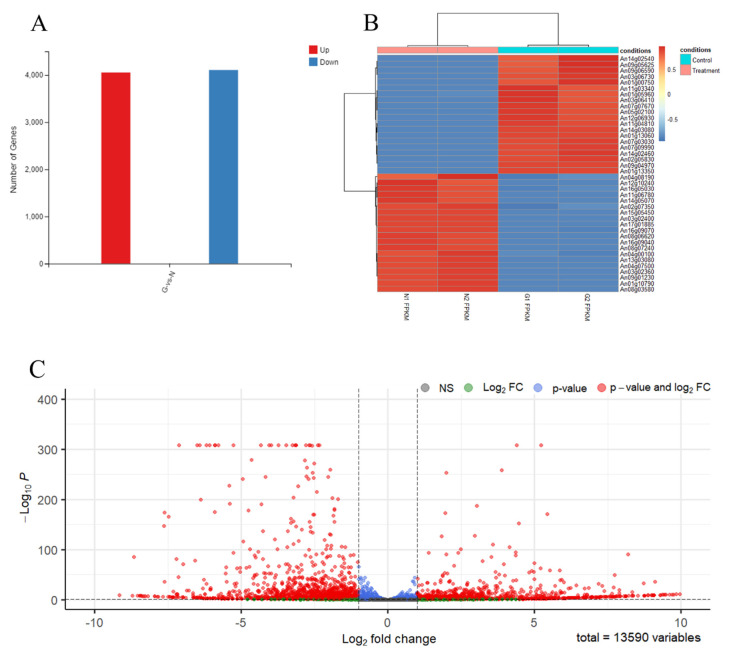
The RNA sequence of the spore−like propagule and the hypha. (**A**) The amount of all up−regulated and down−regulated genes between the spore-like propagule and the hypha. (**B**) The heat map of the 20 highest up−regulated genes and the 20 lowest down−regulated genes between the spore−like propagule and the hypha. |log2(N/G)| > 1, *p* < 0.05. (**C**) The enhanced volcano of 13,590 genes expression between the eight spore-like propagules and the hyphae. |log2(N/G)| > 1, *p* < 0.05. (**D**) Pie chart of the most significantly enriched categories within the up−regulated genes between the spore−like propagule and the hypha. The method was GO and the referenced species was *Aspergillus niger* CBS513.88 (AspGD). (**E**) Pie chart of the most significantly enriched categories within the down−regulated genes between the spore−like propagule and the hypha. The method was GO and the referenced species was *Aspergillus niger* CBS513.88 (AspGD). (**F**) The enriched patways of differentially expressed genes between the spore−like propagule and the hypha. |log2(N/G)| > 1, *p* < 0.05. (**G**) The enhanced volcano of 416 MAPK signaling pathway genes expression between the spore−like propagule and the hypha. |log2(N/G)| > 1, *p* < 0.05. (**H**) The enhanced volcano of 42 conidia-associated genes (CAGs) expression between the spore-like propagule and the hypha. |log2(N/G)| > 1, *p* < 0.05. (**I**) The enhanced volcano of 44 germination-associated genes (GeAGs) expression between the spore-like propagule and the hypha. |log2(N/G)| > 1, *p* < 0.05. (**J**) The enhanced volcano of 37 genes involved in the production of enzymes involved in cell wall sythesis or processing expression between the spore−like propagule and the hypha. |log2(N/G)| > 1, *p* < 0.05.

**Figure 4 jof-08-00679-f004:**
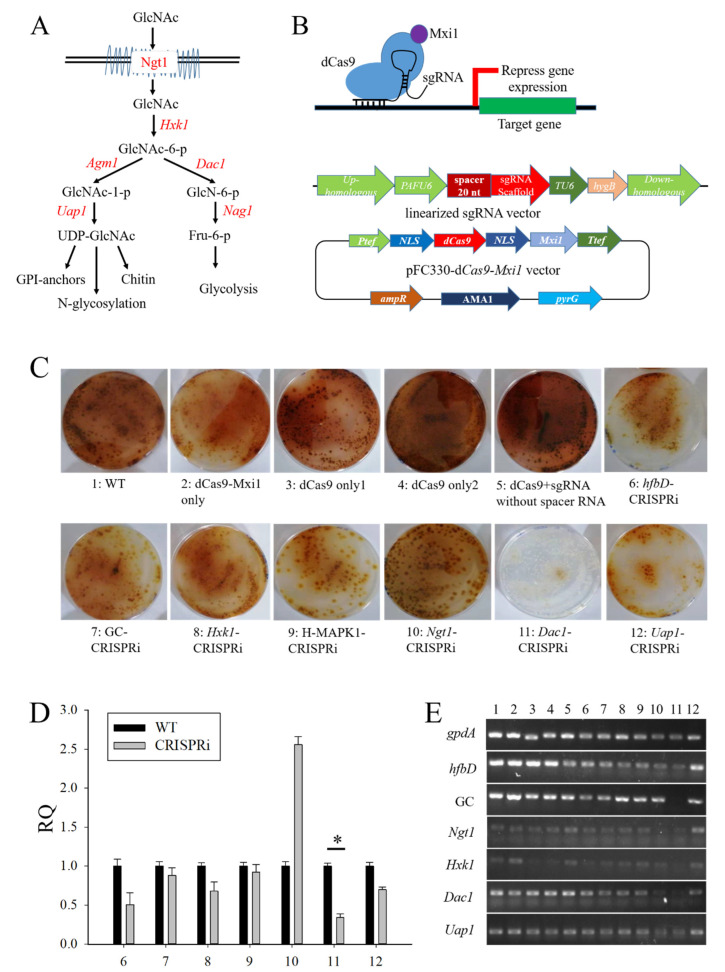
The construction of CRISPRi on conidia and GlcNAc metabolic pathway genes in *Aspergillus niger* SH2. (**A**) The GlcNAc metabolic pathway in Candida albicans. The red font denotes relevant genes. (**B**) Diagram of the CRISPRi working principle and the map of CRISPRi system in this study. In this study, the CRISPRi system consisted of d*Cas9*-*Mxi1* expression cassette and sgRNA expression cassette. The d*Cas9*-*Mxi1* expression cassette was assembled into a pFC330 plasmid. However, the sgRNA expression cassette was inserted into the genome of *Aspergillus niger* SH2 (*aamA* gene locus). (**C**) The phenotype of the *Aspergillus niger* SH2 transformants on NCD plates, where 1 WT means *Aspergillus niger* SH2 wild strain, 2 d*Cas9*-*Mxi1* only means the transformant for which only the pFC330-d*Cas9*-*Mxi1* vector was transferred into *Aspergillus niger* SH2, 3 d*Cas9* only1 means the transformant No. 1 for which only the pFC330-d*Cas9* vector without *Mxi1* was transferred into *Aspergillus niger* SH2, 4 d*Cas9* only2 means the transformant No. 2 for which only the pFC330-d*Cas9* vector without *Mxi1* was transferred into *Aspergillus niger* SH2, 5 d*Cas9* + sgRNA without spacer RNA means the transformant for which the pFC330-d*Cas9*-*Mxi1* vector and linearized sgRNA general vector, without spacer RNA, were simultaneously transferred into *Aspergillus niger* SH2, 6 hfbD-CRISPRi means the transformant for which the pFC330-d*Cas9*-*Mxi1* vector and linearized *hfbD* sgRNA general vector were simultaneously transferred into *Aspergillus niger* SH2, 7 GC-CRISPRi means the transformant for which the pFC330-d*Cas9*-*Mxi1* vector and linearized GC sgRNA general vector were simultaneously transferred into *Aspergillus niger* SH2, 8 Hxk1-CRISPRi means the transformant for which the pFC330-d*Cas9*-*Mxi1* vector and linearized *Hxk1* sgRNA general vector were simultaneously transferred into *Aspergillus niger* SH2, 9 H-MAPK1-CRISPRi means the transformant for which the pFC330-d*Cas9*-*Mxi1* vector and linearized H-MAPK1 sgRNA general vector were simultaneously transferred into *Aspergillus niger* SH2, 10 Ngt1-CRISPRi means the transformant for which the pFC330-d*Cas9*-*Mxi1* vector and linearized Ngt1 sgRNA general vector were simultaneously transferred into *Aspergillus niger* SH2, 11 Dac1-CRISPRi means the transformant for which the pFC330-d*Cas9*-*Mxi1* vector and linearized *Dac1* sgRNA general vector were simultaneously transferred into *Aspergillus niger* SH2, and 12 *Uap1*-CRISPRi means the transformant for which the pFC330-d*Cas9*-*Mxi1* vector and linearized *Uap1* sgRNA general vector were simultaneously transferred into *Aspergillus niger* SH2. (**D**) qPCR analysis of the *Aspergillus niger* SH2 transformants. Samples 6, 7, 8, 9, 10, 11, and 12 are described as per (**C**). (**E**) Semi-quantitative RT-PCR of the *Aspergillus niger* SH2 transformants. Samples 1, 2, 3, 4, 5, 6, 7, 8, 9, 10, 11, and 12 are described as per (**C**). (**F**) The phenotype and micromorphology of the *Aspergillus niger* SH2 transformants on NCD plates. The * means *p* < 0.05.

**Figure 5 jof-08-00679-f005:**
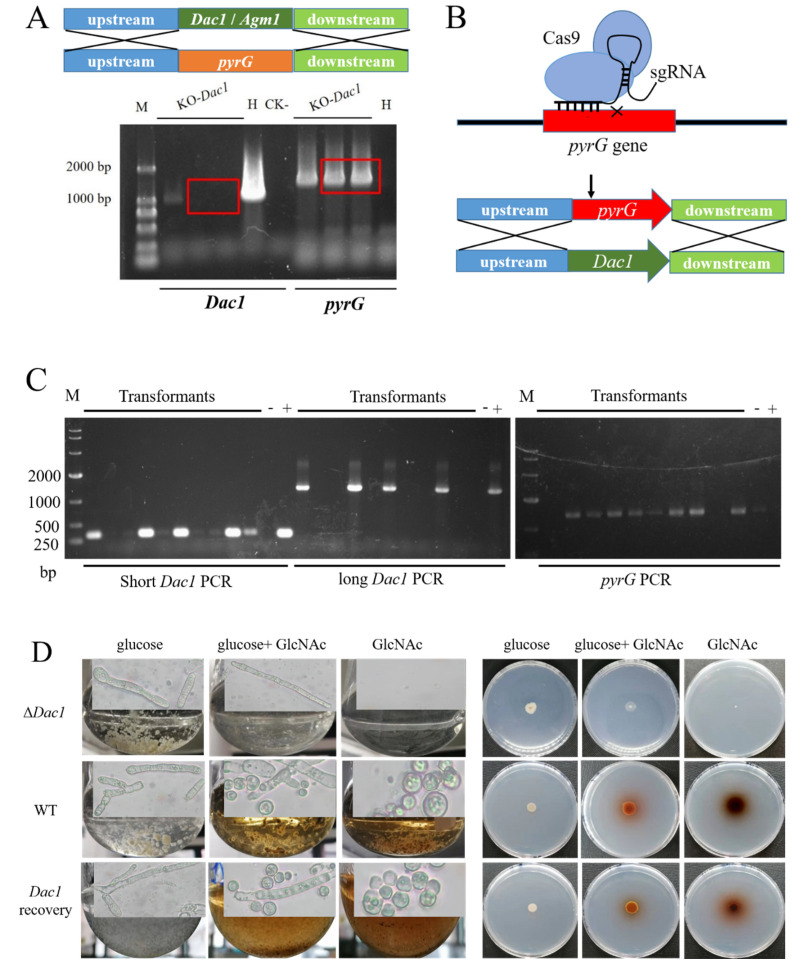
The phenotypes assay of *Dac1* gene knockout and in-situ complementation mutants in *Aspergillus niger* SH2. (**A**) The flowchart of gene knockout and the identification of the gene knockout mutants by PCR. (**B**) One step method was used to recycle the screening tag (*pyrG*) and recover the *Dac1* gene by the CRISPR-Cas9 system. (**C**) The PCR identification of *Dac1* recovery strains. (**D**) The phenotypes and micromorphology assay of *Dac1* gene knockout, in-situ complementation mutants, and wild strain in CD, NGCD and NCD mediums.

## Data Availability

All relevant data generated or analyzed during this study are included in this article.

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
