# Peer review of "A Special Phenotype of Aconidial Aspergillus niger SH2 and Its Mechanism of Formation via CRISPRi"

_jof, 2022, doi:10.3390/jof8070679_

Round 1
Reviewer 1 Report
The present study deeply explores the regulatory pathways of spore-like propagule formation in A. niger SH2, an atypical strain of the species.
Undoubtedly the work has scientific merit and expands the frontier of mycology, the main findings and gains were:
- the development of a functional CRISPRi system and a CRISPR-Cas9-based method to rapidly recycle screening tag (pyrG) and recover the Dac1 gene in A. niger SH2; such systems can serve as a model to search for phenotypic changes in other A. niger strains of industrial importance.
Proof, through the deletion and recovery of the Dac1 gene, of the involvement of the GlcNAc metabolic pathway in the development of spore-like propagules.
-And the conclusion that spore-like propagules are resting conidia going into dormancy, and are more tolerant of environmental stresses. Furthermore, vital activities such as drug resistance, genetic transformation and germination were found in the propagules and systematically investigated by the authors.
The present report appears to be an intermediate study, as it is understood that the authors intend to expand the analysis in future work; it would be interesting to include in this next study the comparison of the SH2 strain with other strains of A. niger with special morphotypes, such as those reported in Frisvad et al. (2014- https://doi.org/10.1371/journal.pone.0094857) and a new cryptic species of A. niger clade, A. vinaceus that is capable of produce sclerotia (Silva et al., 2020- https://doi.org/10.3390/jof6040371).
I consider that the work has objectives of interest that are additive to the area, has an adequate and current methodology, has important results, and provides a clear discussion, therefore, my recommendation to JoF is to accept its publication.
However, I have minor corrections:
Please improve the resolution of all figures, in addition, a cleaner layout of the figures would also make the paper more likable.
Reviewer 2 Report
Dear authors,
The manuscript is a complex one with the potential to be published in the “JoF” journal, but it is recommended to improve some chapters and some sentences. Below are the comments and the recommendations to be implemented point by point. Also, many mistakes are found throughout the manuscript that needs to be corrected (text manuscript, sentences, references, etc.).
1. Line 21 – Correct it “…Dac1 gene. this study can…”.
2. I recommend avoiding using the exact words in the title and the Keywords section.
3. Lines 22 - 35 – Fungi is one of the most used in the industry for enzyme production. I strongly recommend adding this important characteristic. Also, I recommend adding the reference: Martau, G.-A.; Unger, P.; Schneider, R.; Venus, J.; Vodnar, D.C.; Pablo, L.-G. Integration of Solid State and Submerged Fermentations for the Valorization of Organic Municipal Solid Waste. J. Fungi 2021, 7, 766.
4. Line 75 – “The aims of the present study is 1)…” Modify as “The aims of the present study are: 1)…”.
5. Line 100 – “potapo extract (300)” 300 g of infused potato or extract powder? I recommend checking.
6. Figure 1 contains a low resolution, and it is recommended to change it to an acceptable rezolution. Also, on lines 131-132, A. niger needs to be written in italics font.
7. Line 171 – “…and 0.8 M…” 0.8 M or 0.8% ? Please check.
8. Figure 2 – Which plates are for what type of preservation? Mention them in the figure.
9. Line 192 – “…A. niger SH2…” Names of fungi are written in italics format. Also, I recommend checking the whole manuscript for these problems (Lines 257, 269, 270, etc…).
10. Line 518 – “Oakland et al. reported…” Mention the year of publication.
11. Most of the discussions (chapter 4) have been complex discussed but repeat the results obtained in the previous chapter (chapter 3). I strongly recommend avoiding repeating the same argument. It is essential to discuss and compare the results with other articles that have made similar products and the same working methods.
Round 2
Reviewer 2 Report
Dear authors,
The authors have cautiously replied and fully revised the manuscript to the suggestions and comments of the reviewer. Therefore, this work should be recommended for publication in this journal.
